# Mutation of Respiratory Syncytial Virus G Protein’s CX3C Motif Attenuates Infection in Cotton Rats and Primary Human Airway Epithelial Cells

**DOI:** 10.3390/vaccines7030069

**Published:** 2019-07-19

**Authors:** Binh Ha, Tatiana Chirkova, Marina S. Boukhvalova, He Ying Sun, Edward E. Walsh, Christopher S. Anderson, Thomas J. Mariani, Larry J. Anderson

**Affiliations:** 1Pediatric Infectious Diseases, Emory University and Children’s Healthcare of Atlanta, Atlanta, GA 30322, USA; 2Sigmovir Biosystems, Inc. Rockville, MD 20850, USA; 3Department of Medicine, University of Rochester School of Medicine and Department of Medicine, Rochester General Hospital, Rochester, NY 14621, USA; 4Department of Neonatology, Program in Pediatric Molecular and Personalized Medicine, and Department of Microbiology and Immunology, University of Rochester School of Medicine, Rochester, NY 14642, USA

**Keywords:** respiratory syncytial virus, live RSV vaccines, RSV G protein

## Abstract

Despite being a high priority for vaccine development, no vaccine is yet available for respiratory syncytial virus (RSV). A live virus vaccine is the primary type of vaccine being developed for young children. In this report, we describe our studies of infected cotton rats and primary human airway epithelial cells (pHAECs) using an RSV r19F with a mutation in the CX3C chemokine motif in the RSV G protein (CX4C). Through this CX3C motif, RSV binds to the corresponding chemokine receptor, CX3CR1, and this binding contributes to RSV infection of pHAECs and virus induced host responses that contribute to disease. In both the cotton rat and pHAECs, the CX4C mutation decreased virus replication and disease and/or host responses to infection. Thus, this mutation, or other mutations that block binding to CX3CR1, has the potential to improve a live attenuated RSV vaccine by attenuating both infection and disease pathogenesis.

## 1. Introduction

The respiratory syncytial virus (RSV) was first isolated in the 1950s [1,2] and it was quickly recognized as an important cause of acute lower respiratory tract illness in young children. It is now known to cause both upper and lower respiratory tract illnesses including otitis media, bronchitis, bronchiolitis, and pneumonia. The RSV infects most children by two years of age and then causes repeat infections with disease throughout life [3] with compromised cardiac, pulmonary, and immune systems and in older age is associated with serious complications of infection [4,5]. It is estimated that in children <5 years of age, the RSV causes 95,000–150,000 deaths globally and up to 175,000 hospitalizations in the United States and, in older adults, ~14,000 deaths [6,7,8,9]. RSV hospitalizations are also associated with later development of reactive airway disease or asthma [10,11]. Though the burden of RSV has made it a high priority for vaccine and antiviral drug development [12], no vaccine or effective antiviral drug is yet available. Immune prophylaxis is available and effective for preventing serious complications of RSV infection in high-risk young children.

Four potential target populations for RSV vaccines have been identified, infants and young children, older children, pregnant women to protect their infants after birth, and elderly adults [12]. Infants and young children have the highest risk of severe disease and likely benefit the most from a vaccine. The first vaccine, formalin-inactivated RSV with alum adjuvant (FI-RSV) when given to infants and young, who were likely RSV naive children, led to enhanced RSV disease (ERD) with later natural infection [13,14,15,16]. Consequently, vaccine development for young children has been limited to live attenuated RSV or virus vectors expressing RSV proteins. Unfortunately, neither a live attenuated RSV nor a virus vector vaccine has yet been sufficiently safe and effective to move to licensure. Given the history of failed vaccines [17], novel ways to improve live RSV vaccines are needed.

Our data suggest that mutations in the G protein might provide a new way to improve both safety and immunogenicity of a live RSV vaccine. We have shown that the RSV G protein is an important contributor to RSV disease, likely through its CX3C chemokine motif binding to the corresponding chemokine receptor, CX3CR1 [18,19,20,21,22,23,24,25,26]. In the mouse model, blocking G binding to CX3CR1 decreases disease with infection by a mechanism not dependent on a decrease in infection but through an anti-inflammatory-like effect. Importantly, CX3CR1 is also an important receptor for infection of primary human airway epithelial cells (pHAECs) and blocking G binding to CX3CR1 decreases infection of pHAECs [27,28,29]. Since airway epithelial cells are the primary site of infection in humans, mutations in G that prevent this binding should decrease infection and attenuate the virus. In mice, preventing the G-CX3CR1 interaction with anti-G antibodies or the CX4C mutation also improves the later adaptive immune response [25,30]. Thus, in humans, a virus with a mutation in CX3C that blocks the G-CX3CR1 interaction should attenuate RSV by two mechanisms, i.e., decreasing inflammation and infection, and enhancing the adaptive immune response. We previously showed that infection of mice using a virus with the A2 G protein CX3C motif (182CWAIC186) mutated to CX4C (182CWAIAC187) induced substantially less disease with primary infection than the wildtype virus [25].

In this study, we chose to continue evaluation of the CX4C mutation for a live RSV vaccine with infection studies in cotton rats and pHAECs. The cotton rat is more susceptible to RSV, and thus a better infection model than the mouse. Since airway epithelial cells are the primary site of infection in humans, we chose pHAECs to model human infection. In these studies, we focused on the A2 r19F strain of RSV because it has mutations in F that improves its pre-fusion stability [31], and pre-fusion stabilized F induces higher titer of neutralizing antibodies [32,33]. These data show that cotton rat infection with the CX4C virus as compared with the CX3C virus gave a lower RSV titer in the lung and less inflammation. Similarly, in pHAECs, the CX4C virus replicated to a lower level and induced a lower level of cytokines and chemokines associated with inflammation. These studies support the value of the mutating the CX3C motif to improve a live RSV vaccine.

## 2. Materials and Methods

### 2.1. Cotton Rat Studies

#### 2.1.1. Animals

Sixty (60) inbred male and female *Sigmodon hispidus* cotton rats aged 6–8 weeks (Sigmovir Biosystems, Inc., Rockville MD) were maintained and handled under veterinary supervision in accordance with NIH guidelines and the Sigmovir Institutional Animal Care and Use Committee’s approved animal study protocol (IACUC protocol #15). The animals were housed in clear polycarbonate cages individually and provided with standard rodent chow (Harlan #7004) and tap water ad lib. For primary infection studies, naïve adult cotton rats were inoculated intranasally with r19F_CX4C_ or the wildtype r19F virus at 2 × 10^5^ TCID_50_ dose/rat and sacrificed on days 1, 5, and 49 post infection (Table 1). For protection studies, the cotton rats were infected intranasally with the wildtype or mutant virus at 2 × 10^5^ TCID_50_ dose/rat once 7 weeks before being challenged with RSV/A2 (Table 1). For the RSV/A2 virus, a dose of 10^5^ plaque-forming units (PFU) per animal was given once intranasally 7 weeks before being rechallenged with RSV/A2. The control PBS and FI-RSV Lot100 (1:100 dose/animal) were administered twice, at 7 and 4 weeks prior to RSV/A2 challenge, by intramuscular injection (quadriceps) in 0.1 mL. All animals were challenged with 5.0 Log10 PFU RSV/A2 per rat. Retro-orbital sinus bleed was performed for blood collection; for terminal procedures, animals were euthanized by CO_2_ asphyxiation. Table 1 summarizes animal study groups and the associated immunizations and challenge.

#### 2.1.2. Virus Preparation

The prototype A2 strain of RSV (ATCC, Manassas VA) was propagated in HEp-2 cells after serial plaque purification to reduce defective-interfering particles. A stock virus designated as RSV/A2 Lot# 092215 SSM containing approximately 5.0 × 10^8^ PFU/mL in sucrose-stabilizing media and stored at −80 °C for inoculation. This stock of virus has been characterized and validated for upper and lower respiratory tract replication in the cotton rat model. The r19F and r19F_CX4C_ (the r19F_CX4C_ virus was provided by Dr. Marty Moore, formerly at Emory Uiversity and presently with Meissa Vaccines, Inc., South San Francisco, CA) viruses were grown in HEp-2 cells at 0.1 or 0.01 multiplicity of infection (MOI) to reduce defective-interfering particles, purified through a sucrose cushion, and stored at −80 °C [25]. For inoculation, the virus was thawed, diluted to a final concentration with PBS (pH 7.4 *w/o* Ca^2+^ or Mg^2+^), kept on ice, and used within an hour.

#### 2.1.3. Lung and Nose Viral Titration

At the indicated point in time, lung and nose homogenates were clarified by centrifugation and diluted in Eagle’s medium essential medium (EMEM). Confluent HEp-2 monolayers were infected in duplicates with diluted homogenates in 24-well plates. After one-hour of incubation at 37 °C in a 5% CO_2_ incubator, the wells were overlaid with 0.75% methylcellulose medium. After 4 days of incubation, the overlay was removed and the cells fixed with 0.1% crystal violet stain for one hour, rinsed, and air dried. Plaques were counted and the virus titer was expressed as the plaque forming unit per gram of tissue. Viral titers are presented as the geometric mean with standard error for all animals in a given group.

#### 2.1.4. Pulmonary Histopathology

Lungs were removed, inflated with, and immersed in, 10% neutral buffered formalin to their normal volume for 24 h. The lungs were then embedded in paraffin, sectioned, and stained with hematoxylin and eosin (H&E). Four parameters of pulmonary inflammation were evaluated: peribronchiolitis (inflammatory cells infiltration around the bronchioles), perivasculitis (inflammatory cells infiltration around the small blood vessels), interstitial pneumonia (inflammatory cells infiltration and thickening of alveolar walls), and alveolitis (inflammatory cells within the alveolar spaces). Slides were scored blind on a 0–4 scale and subsequently converted to a 0–100% histopathology scale.

#### 2.1.5. RSV Binding IgG Antibodies (ELISA)

Purified RSV/A2 F protein was used as coating antigens in a 96-well plate. The plates were incubated in blocking solution for 1 h at room temperature (RT) and subsequently washed. Serially diluted sera (1:500, 1:2000, 1:8000, and 1:32,000 in duplicates) along with the positive and negative controls were added to the wells and incubated for 1 h at RT. After washing the plates, rabbit anti cotton rat IgG (1:4000) was added to all the wells followed by incubation for 1 h at RT. After washing, the plates were incubated with goat anti rabbit IgG-HRP (1:4000) for 1 h at RT. For development, TMB substrate was added to all the wells and the plates were incubated for 15 min at RT. KPL TMB (3,3’,5,5’- tetramethylbenzidine) peroxidase substrate was added to all the wells and the plates were incubated for 15 min at RT. KPL TMBstop solution was added to all the wells and optical density (OD) at 450 nm was recorded. Geometric mean of the OD_450_ was calculated from the duplicates.

#### 2.1.6. RSV Neutralizing Antibody Assay

Heat inactivated sera were diluted 1:10 with EMEM and serially diluted further 1:4. Diluted sera were incubated with RSV/A2 (25–50 PFU) for 1 h at RT followed by inoculation in duplicates onto confluent HEp-2 monolayers in 24-well plates for 1 h at 37 °C in a 5% CO_2_ incubator. The wells were then overlaid with 0.75% methylcellulose medium. After 4 days of incubation, the overlays were removed and the cells were fixed and stained with 0.1% crystal violet for 1 h, then rinsed and air dried. The corresponding reciprocal neutralizing antibody titers were determined at the 60% reduction endpoint of the virus control using the statistical program “plqrd.manual.entry”. The geometric means ± SEM for all animals in a group at any given time were calculated.

#### 2.1.7. Real-Time Polymerase Chain Reaction (PCR)

Total RNA was extracted from homogenized lung or nasal tissue using RNeasy purification kit (QIAGEN). To prepare cDNA, 1 μg of total RNA was used with Super Script II RT (Invitrogen) and 1 μL of oligo dT primer (Invitrogen). Bio-Rad iQ SYBR Green supermix was used for real-time PCR reactions at 25 μL of final volume and 0.5 μM of primer concentrations. Reactions were set up in duplicates in 96-well plates and the amplifications were performed on a Bio-Rad iCycler as following: 1 cycle at 95 °C for 3 min, followed by 40 cycles at 95 °C for 10 sec, 60 °C for 10 sec, and 72 °C for 15 sec. The baseline cycles and cycle threshold (C_T_) were calculated by the iQ5 software in the PCR Base Line Subtracted Curve Fit mode. Relative quantitation of DNA was applied to all samples and the standard curves were graphed using serially diluted cDNA sample most enriched (e.g., lungs from day 4 post infection). The C_T_ values were plotted against log10 cDNA dilution factor. These curves were used to convert the C_T_ values obtained for different samples to relative expression units, which were then normalized to the level of the housekeeping gene β-actin mRNA from the same sample. The mRNA levels were expressed as the geometric mean ± SEM for all animals in a group at any given time.

### 2.2. pHAECs Studies

#### 2.2.1. Cells

The pHAECs from healthy adult patients were kindly provided by Dr. C.U. Cotton (Case Western Reserve University), expanded, and cultured as described [34,35]. Briefly, cells were plated on a layer of mitomycin C (Sigma) treated 3T3 mouse fibroblast feeder cells and grown until 70% confluency in the 1:3 mixture of Ham’s F12/DMEM media (HyClone) supplemented with 5% FBS (Sigma), 24 µg/mL adenine (Sigma), 0.4 µg/mL hydrocortisone (Sigma), 5 µg/mL Insulin (Sigma), 10 ng/mL EGF (Sigma), 8.4 ng/mL Cholera toxin, and 10 µM ROCK1 inhibitor Y-2763 (Selleck Chemical LLC). After expansion, cells (passage 3 or 4) were plated on Costar transwell inserts (Corning Inc., Corning, NY), grown until confluency, then transferred to air-liquid interface where cells were maintained in 1:1 mixture of Ham’s F12/DMEM media supplemented with 2% Ultroser G (Pall Biosepra, SA, Cergy-Sainte-Christophe, France) for 3–4 weeks until they were differentiated. Differentiation was confirmed by transepithelial resistance measurements >500 ohms, and flow cytometry staining for FoxJ1 (eBioscience) and acetylated α-tubulin (Life Technologies).

#### 2.2.2. RSV Inoculation

The differentiated pHAECs were inoculated with RSV MOI of 0.1 or 0.3 as determined by infectivity titration in HEp-2 cells. The pHAECs were washed with PBS and virus in PBS or PBS alone was added to the apical surface of the cells and incubated for 2 h at 37 °C, the virus inoculum aspirated, the apical surface washed with PBS and fresh media added to the basolateral compartment. The RSV-infected pHAECs were incubated for 2, 4, or 6 days at 37 °C and 5% CO_2,_ and then cells were collected by replacing the lower chamber media with trypsin-0.05% EDTA (Gibco) to release the cells from the insert. After collection, the released cells were washed twice with sterile PBS.

#### 2.2.3. Real-Time PCR

Total RNA was extracted and purified from the cells using a Qiagen RNeasy kit (QIAGEN). The RNA was reverse transcribed into cDNA using an iScript^TM^ cDNA synthesis kit (Bio-Rad) following the manufacturer’s instruction. The quantitative PCR was carried out on a 7500 Fast Real-time PCR system (Applied Biosystems) using Power SYBR Green PCR master mix (Applied Biosystems). The C_T_ values were normalized using control β-actin C_T_ values from the same samples. The RSV matrix M gene primers and amplification cycles were described previously [25]. Other primer pairs used were: β-actin, forward 5′-CAC CAA CTG GGA CGA CAT-3′, reverse 5′- ACA GCC TGG ATA GCA ACG-3′; Muc5AC: forward 5′-GGA GGT CCC ACT TCT CAAC-3, reverse 5′-CTT CAG GCA GGT CTC GCT G-3′; Muc5B: forward 5′-GCC TAC GAG GAC TTC AAC GTC-3′, reverse 5′-CCT TGA TGA CAA CAC GGG TGA-3′; IL6: forward 5′- CCT GAA CCT TCCA AAG ATG GC-3′, reverse 5′- TTC ACC AGG CAA GTC TCC TCA-3′; CX3CL1: forward 5′- CGC GCA ATC ATC TTG GAG AC-3′, reverse 5′- CAT CGC GTC CTT GAC CCA T-3′; CCL7 forward 5′- AAA CCT CCA ATT CTC ATG TGG AA-3′, reverse 5′- CAG AAG TGC TGC AGA GGC TTT-3′.

#### 2.2.4. Flow Cytometry

Cells were transferred to a 96-well plate, washed with staining buffer (PBS containing 0.5% bovine serum albumin (BSA) and 0.05% sodium azide), fixed with BD FACS lysing solution, permeabilized with BD FACS permeabilization solution 2, blocked with PBS containing 0.5% BSA, incubated with human anti RSV F monoclonal antibody (kindly provided by MedImmune, Gaithersburg, MD), and Alexa Fluor^®^ 488 labeled goat anti human IgG (H+L) antibody (Invitrogen). Flow cytometry was performed using 4-laser BD LSF-II (BD Biosciences) and the results analyzed with FlowJo software (Tree Star Inc., Ashland, OR).

#### 2.2.5. Statistical Analyses

Unless otherwise indicated, different groups were compared by nonparametric ANOVA (Kruskal–Wallis), Mann–Whitney, Wilcoxon rank sum test or Wilcoxon matched pairs test for comparison of treatments within the same group. A *p* value of <0.05 was considered statistically significant.

## 3. Results

### 3.1. Significant Reduction of Viral Load in Lungs and Nose of r19F_CX4C_-Infected Cotton Rats

To investigate the effects of mutating the CX3C motif in G (182CWAIC186) to CX4C (182CWAIAC187) on infection and disease pathogenesis in cotton rats, we compared the r19F to the r19F_CX4C_ virus. We included infection with the A2 virus to provide a reference point for attenuation to other cotton rat studies with A2 and r19F where the viruses were not purified through a sucrose cushion [32,36,37]. The A2 virus also provides another comparison to the FI-RSV vaccination which was included to provide a reference for replication and disease associated with enhanced disease. Sixty cotton rats were divided into six groups as shown in Table 1. All three viruses replicated in the lung and nose after primary infection (Figure 1) (*n* = 5 per group). The r19F_CX4C_-infected animals had a lower lung and nasal RSV titers relative to r19F-infected animals on both day one and day five post infection. We believe infectious virus detected in lung and nasal tissue on day one represents virus replication since RSV quickly loses titer over hours in the environment [38]. The maximum difference in lung virus titer was on day five post infection when r19F_CX4C_-infected animals had 0.9 Log_10_ less virus (*p* < 0.01) (Figure 1). Moreover, the r19F_CX4C_-infected animals had 1.1 and 0.8 Log_10_ less virus in nasal tissue on day one and day five post infection, respectively, (*p* < 0.01) (Figure 1). These data show that the CX4C mutation attenuated virus replication in the cotton rat. Animals were then challenged with RSV/A2 virus at day 49 post infection and lung and nasal tissues were harvested five days later. No virus was detected in the lung or nose tissue of animals previously infected with r19F, r19F_CX4C_, or A2 (data not shown). In contrast, PBS -immunized or FI-RSV-immunized animals challenged with RSV/A2 had 5 Log_10_ and 3.6 Log_10_ pfu RSV, respectively, in the lung and both groups had 5.6 Log_10_ RSV in the nose tissue at day five post challenge. Thus, r19F_CX4C_ prevented RSV infection of the lung and nose in challenged animals as effectively as A2 and r19F viruses.

### 3.2. r19F_CX4C_ Virus Induces Less Pulmonary Inflammation in Cotton Rats

We next evaluated lung pathology at day 5 post primary and subsequent RSV/A2 challenge infection. For these studies we stained lung tissue with hematoxylin and eosin (H&E) and evaluated four indicators of lung inflammation, i.e., peribronchiolitis (inflammatory cell infiltration around the bronchioles), perivasculitis (inflammatory cell infiltration around blood vessels), interstitial pneumonia (inflammatory cell infiltration and thickening of alveolar walls), and alveolitis (inflammatory cells within the alveolar spaces). These indicators were assessed blinded to the treatment status of the animals. After primary infection compared to mock control infection, r19F- and r19F_CX4C_-infected animals had increased levels of inflammation in the lungs on day 5 post primary infection (Figure 2a,b) at the time of peak virus titers in lungs and nose. r19F_CX4C_-infected, compared to r19F-infected animals, had less inflammation for all indicators which was significant for all combined and individually for interstitial inflammation and alveolitis. Primary infection with A2, r19F and r19F_CX4C_ infections were associated with similar levels of some inflammation on day 5 after challenge with RSV/A2 (Figure 2b). All three viruses primed for much less inflammation after challenge than animals challenged after FI-RSV vaccination (Figure 2b). These differences were significant (*p* < 0.05) for 2 or more parameters for each virus, i.e., A2 for perivasculitis, interstitial pneumonia, and alveolitis; r19F for interstitial pneumonia and alveolitis; and r19F_CX4C_ for peribronchiolitis and perivasculitis. These data support the lung pathology findings in earlier studies in BALB/c mice that the CX4C mutation decreases some parameters of lung inflammation and that r19F may induce more disease than A2.

### 3.3. r19F and r19F_CX4C_ Viruses Comparably Induce Binding and Neutralizing Antibodies

To ascertain the ability of r19F and r19F_CX4C_ viruses to induce RSV-specific antibodies, we determined serum binding antibodies to immobilized purified full-length F protein by ELISA and neutralizing antibodies by plaque assay in serum specimens collected on days 0, 28, and 49 post infection. Primary infection with A2, r19F, and r19F_CX4C_ induced comparable IgG anti-F protein antibodies at day 28 and 49 post infection. The peak titer of antibodies was at day 28 post infection (Figure 3A). FI-RSV-vaccination, however, induced lower levels of anti-F protein antibodies at day 28 and 49 post vaccination. We also determined serum neutralizing antibodies at day 28 and 49 post vaccination by plaque reduction assay. RSV/A2-vaccinated animals produced the highest neutralizing antibody titer at day 28 and 49 post vaccination that were 3–4 Log_2_ higher than the titers in animals vaccinated with RSV r19F or r19F_CX4C_ (Figure 3B). As expected, FI-RSV-vaccinated animals failed to induce neutralizing antibodies. Our data demonstrate that, despite replicating to a lower level, the r19F_CX4C_ virus induced binding and neutralizing antibodies to levels comparable to wildtype r19F though to lower levels than those induced by RSV/A2 infection. 

### 3.4. r19F_CX4C_ Virus Induces Less Inflammatory Cytokine/Chemokine mRNAs in Cotton Rats

To investigate differences in the host inflammatory response associated with the CX4C mutation, we determined transcripts for cytokines, chemokines, and antiviral proteins (e.g., MX1). For these host response experiments, total mRNA from homogenized lungs were converted to cDNA and tested by real-time PCR as described in Materials and Methods. Results show that, compared to r19F-infected animals, r19F_CX4C_-infected animals produced significantly lower levels of cytokines/chemokines (IFN-γ, MCP-1, and TNF-α or an antiviral protein (MX1) at day five post infection (Figure 4). The RSV/A2- compared to r19F-infected animals also had significantly lower levels of these lung mRNAs at day five post primary infection. The MX1 expression is controlled by Type I and III interferons, and it is known to inhibit negative-strand RNA viruses [39] It has been shown to increase in RSV infection of a human immortalized lung cell line, certain of its alleles associated with severity of RSV disease, and its expression negatively associated with RSV replication in pHAECs [40,41,42]. In RSV/A2 challenge studies, the mRNA levels were not significantly different among animals previously infected with A2, r19F or r19F_CX4C_ with the exception that r19F_CX4C_ infected animals had lower levels of MX1 mRNA than those from r19F infected animals (*p* < 0.01). The decreases in these mRNAs with the CX4C virus could result from lower levels of virus replication and/or not binding to CX3CR1.

### 3.5. r19F_CX4C_ Virus Is Less Infectious in Primary Human Airway Epithelial Cells

Since disruption of the CX3C motif in the r19F_CX4C_ virus was associated with decreased infectivity and disease in the cotton rat as well as the BALB/c mouse in an earlier study [25], we investigated the effect of this mutation in the pHAEC model of human infection. In these experiments, we infected pHAECs derived from three different patients with r19F, r19F_CX4C_, or mock and harvested cells and supernatants at day 2, 4, or 6 post infection. Results show that consistent with the cotton rat studies, the percentage of RSV-infected cells as determined by flow cytometry, from two independent experiments showed consistently fewer RSV positive pHAECs with the r19F_CX4C_ virus (Figure 5). The ratio of positive cells for r19F/ r19F_CX4C_ ranged from 2 to 6 on day two, 3 to 12 on day four, and 2 to 11 on day six post infection. The mean ratio for the three patients for both experiments was 3.48 for day two, 5.8 for day four, and 5.18 for day six post infection. PCR for RSV RNA show an average of 2.8-fold less RNA on day two and 1.8-fold less RNA on day four in r19F_CX4C_- compared to r19F-infected cells (Figure 6). The replication data show that r19F_CX4C_ virus is less effective than r19F at infecting pHAECs, which is consistent with the cotton rat data and earlier studies showing the effect of the CX4C mutation on replication of RSV/A2 and CX3CR1 as an important RSV receptor in pHAECs [27,28]. 

### 3.6. pHAECs Response to Infection

To examine the pHAECs response to infection by the two viruses, we determined the mRNA levels for RSV M as well as those of IL-6, CX3CL1, CCL7 (MCP-3), Muc5AC, and Muc5B. We studied these transcripts because they showed differences between the CX3C and CX4C viruses in preliminary studies of the pHAEC transcriptome response to RSV infection. Here, our results show that r19F_CX4C_ infected pHAECs had significantly less RSV M mRNA levels at both day two and day four post infection as compared with those from r19F infected cells (Figure 6). No significant differences were noted between r19F_CX4C_ and r19F infected pHAECs of CX3CL1 and IL-6 mRNAs, although these were less in r19F_CX4C_ infected cells at both day two and four post infection. Interestingly, Muc5AC, which is associated with RSV infection, showed a significantly higher expression of mRNA in r19F-infected cells as compared with r19F_CX4C_ at day four post infection (Figure 6). However, we observed a higher level of Muc5B mRNA levels with r19F_CX4C_ infected cells at day four post infection although the difference was not significant. Similarly, we saw a significant increase of CCL7 mRNA levels in r19F_CX4C_ infected pHAECs as compared with those from r19F infected cells (Figure 6). This agrees with our findings in previous studies where we observed a decrease of MCP-1 expression, which is closely related to CCL7 (MCP-3) by location on chromosome 17 and function, by cells infected with CX4C RSV/A2 compared to CX3C RSV/A2 [27]. These data agree with the findings from the cotton rat study and suggest that the r19F_CX4C_ virus is attenuated for replication and disease associated inflammatory responses that likely apply to humans.

## 4. Discussion

This study was designed to investigate the potential role of a CX3C-to-CX4C mutation in the G protein in live attenuated RSV vaccine. We hypothesized that this mutation would help address two challenges in developing live attenuated RSV vaccines for young children, safety and efficacy. Our earlier studies in mice suggest this mutation might both decrease disease associated with virus replication and virus induced host inflammatory responses and enhance adaptive immune responses otherwise altered by the G protein. The data from the present study support that the mutation improves safety but do not show that it enhances adaptive immune responses. First, the r19F_CX4C_ virus replicated to significantly lower levels in the lung and nose after primary infection in the cotton rat similar to the decrease in lungs seen in earlier BALB/c mouse studies [25]. In pHAECs, the decrease in infection as indicated by percent RSV positive cells at day four and day six post infection for the three pHAECs was substantially greater than the decrease seen in the cotton rat. This greater decrease in pHAECs is not surprising given the importance of CX3CR1 as a receptor in pHAECs [27,28]. The role of CX3CR1 in RSV infection of the cotton rat is unknown. The decrease with r19F_CX4C_ as compared with its parent r19F virus in pHAECs seen in the present study is similar to that seen in our previous studies which compared replication of A2_CX4C_ virus to its parent A2 virus [27,28].

Similar to the mouse [31], in the cotton rat we found that primary r19F virus infection induced more inflammation than the A2 virus. The r19F_CX4C_ virus effectively reversed this increase with primary r19F infection as indicated by interstitial inflammation and alveolitis and levels of MX1, TNF-α, INF-g, and MCP-1 mRNA in lung tissue. It did not, however, decrease lung inflammation or levels of mRNA below levels seen with primary A2 infection. Since the A2 virus has required attenuation for vaccine development, the CX4C mutation in the r19F virus may, by itself, not be sufficiently attenuated for a live attenuated RSV vaccine. The pre-fusion F in r19F is partially stabilized which could be advantageous if, as suggested by other studies, it might induce higher titers of neutralizing antibodies [33]. The present study suggests that a virus that has the pre-fusion F stabilizing mutations without mutations associated with increased lung inflammation and disease would be preferred. It is not known, however, if the mutations associated with these two phenotypes are distinct.

In humans, the CX4C mutation may be more attenuating since the decrease in virus replication is greater than that seen in cotton rats. Infection of pHAECs also showed a decrease in the number of mRNAs associated with host responses to infection with the r19F_CX4C_ virus as compared with the r19F virus. This decrease may be related to a decrease in replication with the CX4C mutation and not its effect on binding to CX3CR1. The decrease in MUC5AC levels observed in r19F_CX4C_ virus infected pHAECs as compared with that in r19F virus infected cells, however, appears more than the decrease in virus replication and is accompanied by an increase in MUC5B levels. An increased level of MUC5AC levels with r19F infection has been described in mice [25,31] and associated with increased severity of RSV-infected infants [43]. The increase in CCL7 with the r19F_CX4C_ as compared with r19F infection of pHAECs is also of interest. In a previous study [27], we noticed a relative increase in MCP-1 (closely related to CCL7) in pHAEC cultures infected with CX4C-mutated RSV A2 virus. Similarity between productions of MCP-1/MCP-3 in pHAEC infected with CX4C mutated A2 and r19F viruses suggests that CX3C motif downregulates these cytokines and monocyte/macrophage activation. This effect seems to be CX3C-restricted and does not depend on F protein interaction with cells. Additionally, in a previous study with A549 cells and human peripheral blood mononuclear cells with wild-type and CX4C-mutated A2, we noted the CX3C motif also downregulated monocyte functions [44].

We considered the partial stabilization of pre-fusion F in r19F to be an attractive feature of this virus because pre-fusion stabilized F is associated with induction of higher titers of neutralizing antibodies [33]. In the present cotton rat study, the r19F virus induced lower rather than higher titers of neutralizing antibodies than the A2 virus. The comparison of the r19F and r19F_CX4C_ with A2 may be confounded by differences in virus preparation and inoculum. The r19F and r19F_CX4C_ viruses were purified through a sucrose cushion while the A2 virus was not. The r19F and r19F_CX4C_ inoculum were also higher, 2 x 10^5^ pfu, as compared with 10^5^ pfu per animal for A2. We are uncertain what effects these differences might have had on our results. In the BALB/c mouse, we noted an increase titer of binding and neutralizing antibodies with the CX4C virus as compared with the parent virus for both 19F and A2 [30]. In the present cotton rat study, although the CX4C virus replicated to lower levels, it induced binding and neutralizing antibodies similar to those seen in the parent r19F virus. Unlike the BALB/c mouse studies, however, the titers were not higher than those induced by the parent virus.

A recent report raised a concern that the CX4C mutation alters immunogenicity of this region of G leading to a substantial decrease in induction of neutralizing antibodies and less protection from RSV challenge in hamsters [45]. In that study, various forms of the G protein are expressed in a human parainfluenza virus 3 vector. In our BALB/c mouse studies with live RSV with and without the CX4C mutation [25], we did see some increase in induction of neutralizing antibodies and similar levels of protection as compared with the wild-type virus. However, we did not include complement in our neutralization assay, therefore, G neutralizing antibodies were not detected. We did, however, see similar level of antibodies against a peptide from the central conserved domain of G after infection with the CX4C and wildtype viruses. Differences between G expressed by a virus vector without F and G expressed during RSV infection with F might explain these differences. This recent report does show that mutations to block binding to CX3CR1 need to be assessed for altered immunogenicity.

## 5. Conclusions

In summary, the CX4C mutation is a promising way to improve the safety of a live attenuated virus through a decrease in virus replication but, likely, more importantly by decreasing disease pathogenesis by a mechanism that is independent of the level of virus replication. Directly altering disease pathogenesis should provide one way to maintain immunogenicity without compromising safety. Though the r19F_CX4C_ virus may be too pathogenic to pursue as a vaccine candidate and the CX4C mutation may not be optimal, mutating the G protein to block binding to CX3CR1 is a promising strategy to improve safety and possibly efficacy of live attenuated RSV vaccines. Since antibodies against G have both an antiviral and anti-inflammatory effect [22,23,24], the G antibody response will likely add to vaccine efficacy and be worth maintaining in a vaccine.

## Figures and Tables

**Figure 1 vaccines-07-00069-f001:**
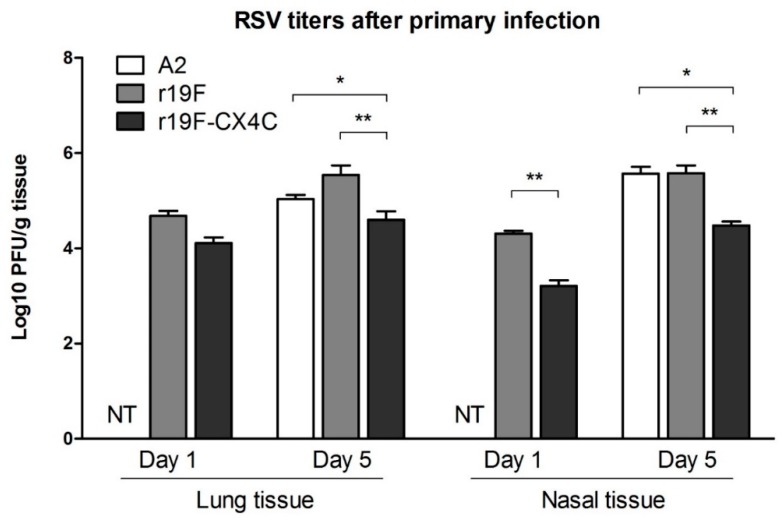
Respiratory syncytial virus (RSV) titers in the lung and nose after primary infection of cotton rats. Five female and/or male cotton rats (6–8 weeks) were infected intranasally with 10^5^ PFU of A2 or 2 × 10^5^ PFU of r19F or 2 × 10^5^ PFU of r19F_CX4C_ in 0.1 mL volume. Animals were sacrificed on the indicated day post infection and lung and nose homogenates used to infect HEp2 monolayers in duplicates for plaque assay infectivity titration. PFUs were determined per gram of tissue. Data are means ± SEMs. Statistical significance is indicated: **, *p* < 0.01 by nonparametric ANOVA (Kruskal–Wallis) and Mann–Whitney test. Note the lung and nose virus titer after A2 primary infection was only tested on day 5 and not on day 1 (NT, not tested).

**Figure 2 vaccines-07-00069-f002:**
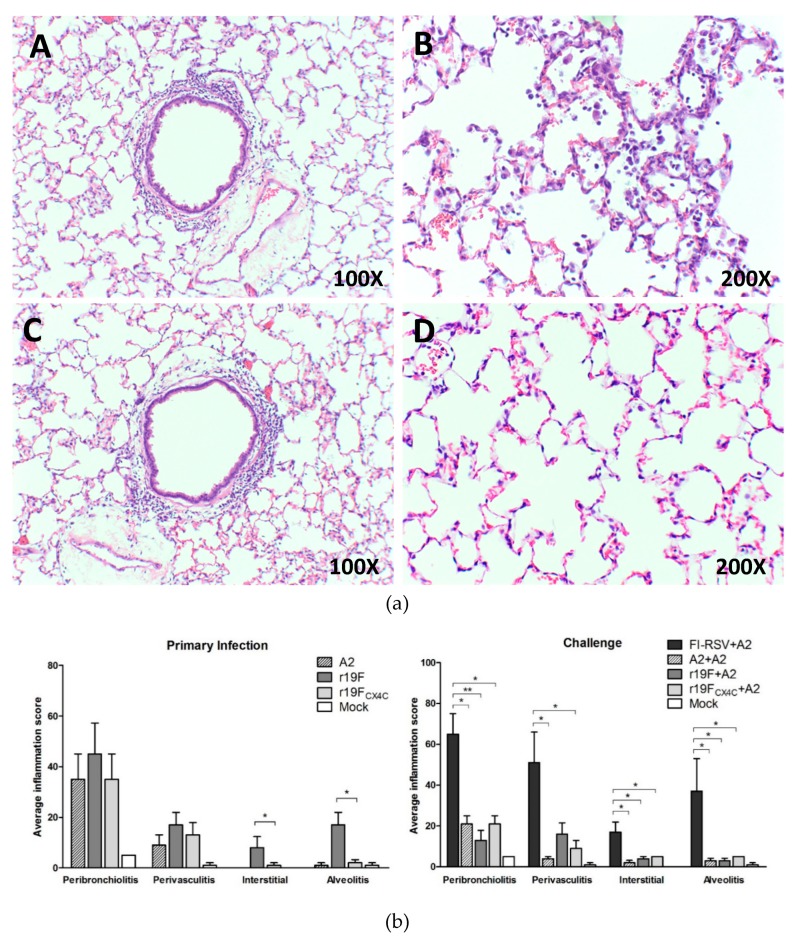
Lung histopathology 5 days after primary infection or challenge of cotton rats. Five female and/or male cotton rats (6–8 weeks) per group were infected intranasally with the 10^5^ PFU of A2 or 2 × 10^5^ PFU of r19F or 2 × 10^5^ PFU of r19F_CX4C_ in 0.1mL volume. Animals were sacrificed and lungs collected, fixed, and stained with hematoxylin and eosin. (**a**) Hematoxyline and eosin (H&E) staining of cotton rats lungs after primary infection with r19F (**A,B**) or r19F_CX4C_ (**C,D**). Original magnifications are shown in the lower right corner of each panel. (**b**) Lung pathology scores after primary infection or A2 challenge. A total of four parameters were evaluated: peribronchiolitis (inflammatory cell infiltration around the bronchioles), perivasculitis (inflammatory cell infiltration around the small blood vessels), interstitial pneumonia (inflammatory cell infiltration and thickening of alveolar walls), and alveolitis (cells within the alveolar spaces). The slides were scored on a 0–4 scale and subsequently converted to a 0–100% histopathology scale. Note that no (0) interstitial (interstitial pneumonia) was seen on day 5 after primary infection with A2 or mock. Statistical significance is indicated: *, *p* < 0.05; **, *p* < 0.01 by nonparametric ANOVA (Kruskal–Wallis) and Mann–Whitney test.

**Figure 3 vaccines-07-00069-f003:**
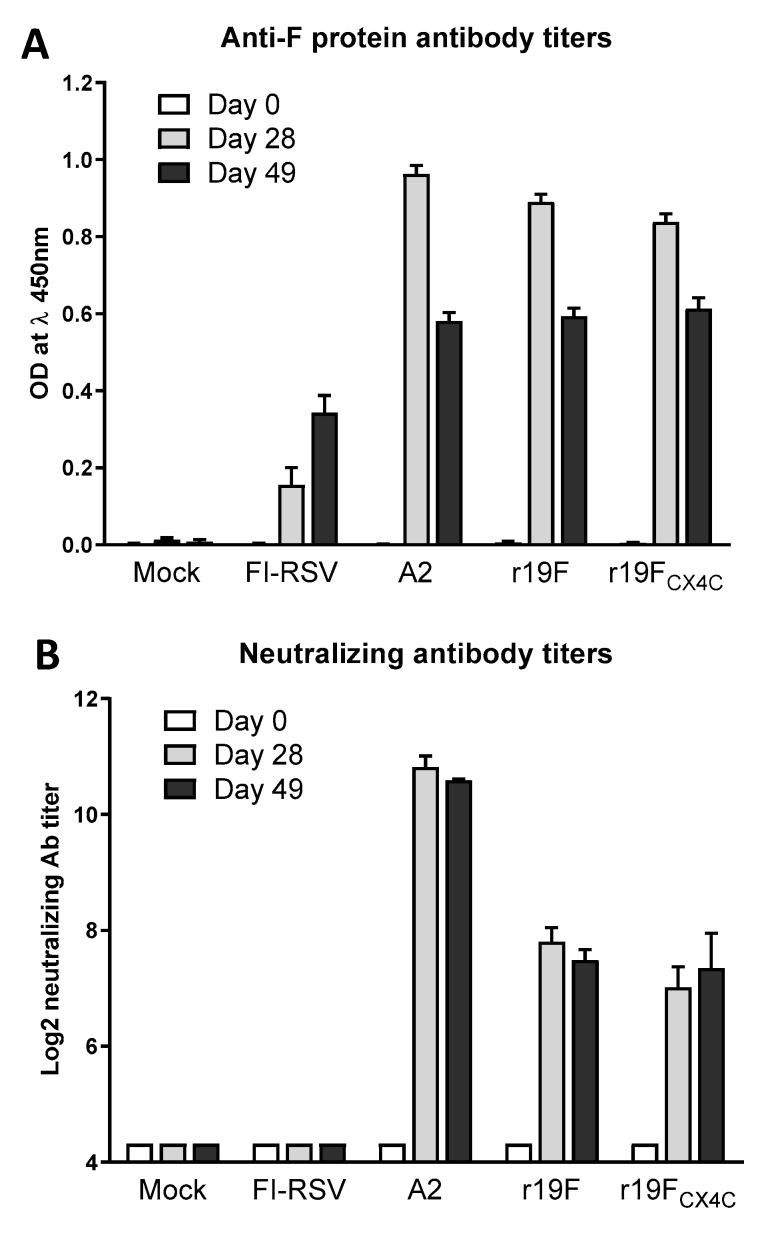
Anti-F protein and neutralizing RSV antibodies. Blood was collected at day 0, 28, and 49 post vaccination/infection. (**A**) The anti-F protein IgG binding antibody was determined by ELISA as described in the materials and methods. The anti-F antibody response is indicated by absorbance for F minus absorbance at a 1:500 dilution. (**B**) The neutralizing antibodies were determined by a 60% plaque reduction assay. The serum specimens were heat inactivated and incubated with RSV A2 (25–50 PFU) at serial 4-fold dilutions beginning at a 1:10 dilution followed by inoculation in duplicates onto confluent HEp-2 monolayers in 24-well plates as described in materials and methods. The reciprocal neutralizing antibody titer is the highest titer with >60% reduction in plaque number as compared with the virus control.

**Figure 4 vaccines-07-00069-f004:**
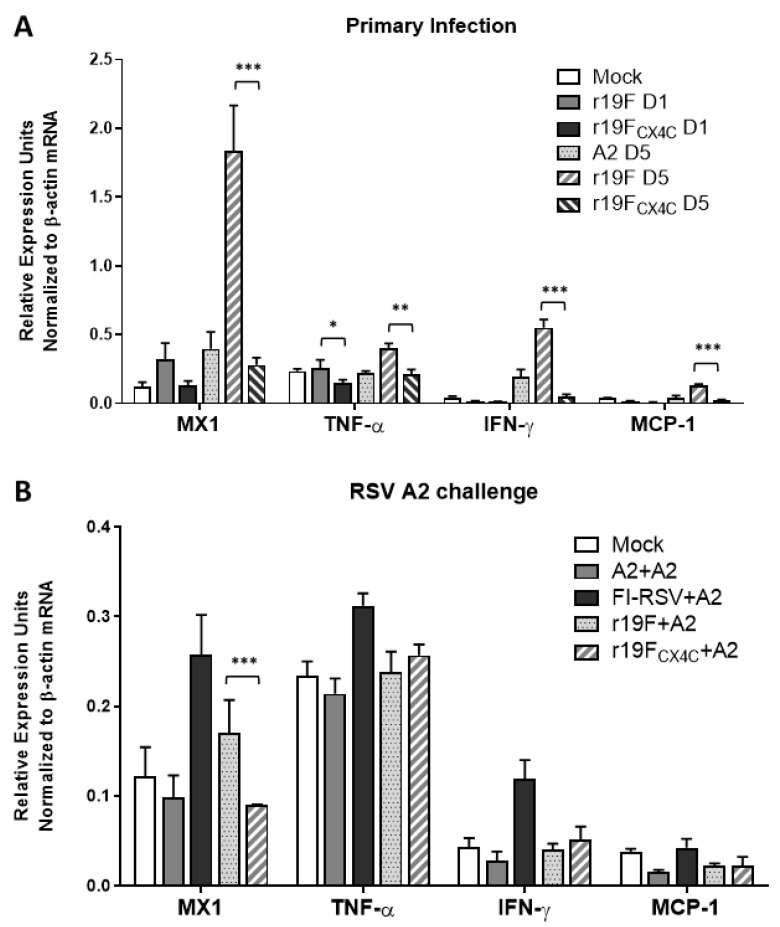
Cytokine/chemokine mRNA in lungs after primary infection (**A**) and day 5 post challenge (**B**) of cotton rats. RNAs were extracted from lung homogenates of infected animals at indicated time and reverse transcribed to cDNA. RT-PCR was performed in duplicates, and baseline cycles and cycle threshold (C_T_) were calculated. Relative quantitation of DNA was determined based on standard curve constructed by serially diluted cDNA of lungs from day 4 post infection of FI-RSV-immunized animals. C_T_ values were shown as relative expression units normalized to the level of β-actin mRNA from same sample. Bars show mean values of mRNA ± SEM for all animals within a group. Statistical significance is indicated: *, *p* < 0.05; **, *p* ≤ 0.01; ***, *p* ≤ 0.001.

**Figure 5 vaccines-07-00069-f005:**
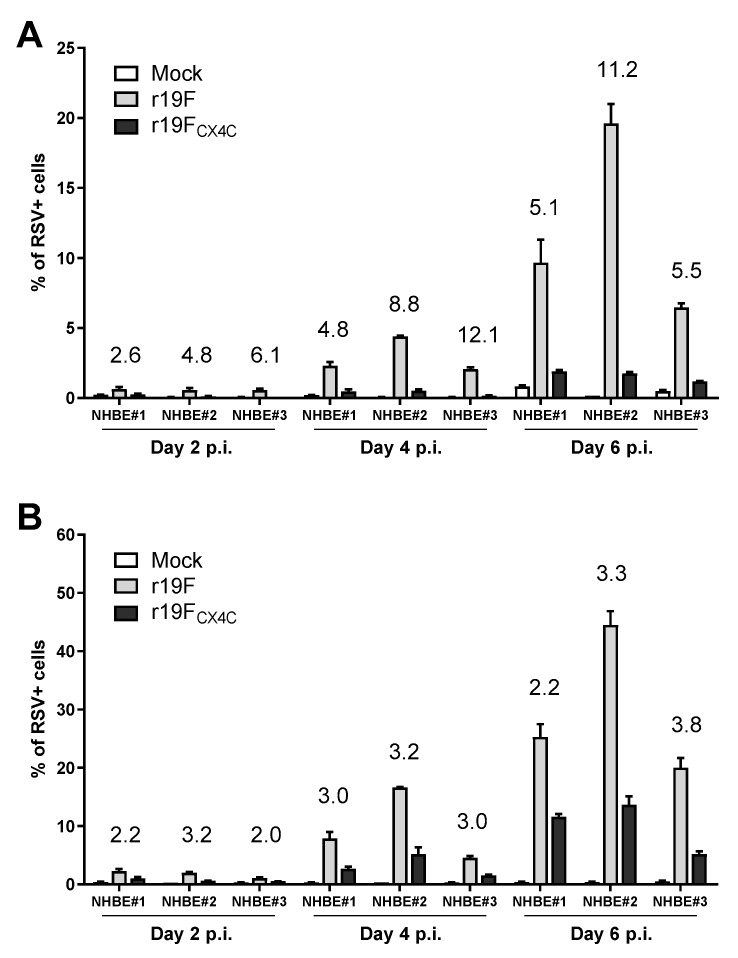
Infectivity of r19F and r19F_CX4C_ viruses in pHAECs. Differentiated pHAECs from three adult donors were inoculated with RSV at multiplicity of infection of 0.1 or 0.3. the r19F or r19F_CX4C_ virus in PBS or PBS alone (mock), was added to the apical surface of the cells and incubated for 2 h at 37 °C. Cells were incubated for 2, 4, or 6 days at 37 °C and 5% CO_2_ and then collected with trypsin-0.05% EDTA. Cells were transferred to a 96-well plate, washed, fixed, permeabilized, and blocked with PBS containing 0.5% BSA. They were then incubated with human anti RSV F monoclonal antibody followed by Alexa Fluor^®^ 488 labeled goat anti human IgG (H+L) antibody incubation. Labeled cells were detected by flow cytometry and results analyzed by FlowJo. Data are expressed as r19F/r19F_CX4C_ ratios. (**A**) Experiment 1. (**B**) Experiment 2. NHBE#1, NHBE#2, NHBE#3—primary human bronchial epithelial cultures from donors 1, 2, 3, respectively. The differences between the r19F and r19F_CX4C_ (ratio is given above the bar) are significant at *p* < 0.01 for day 2, *p* < 0.05 for day 4, and *p* < 0.01 for day 6 for mean values across all three pHAEC specimens for the two independent experiments by the paired t-test.

**Figure 6 vaccines-07-00069-f006:**
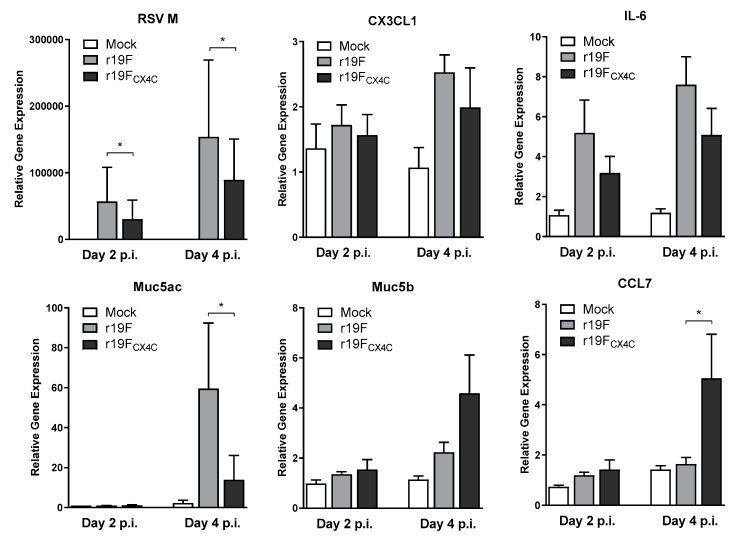
RSV RNA and cellular response mRNAs. The differentiated pHAECs from three adult donors were inoculated with RSV at a multiplicity of infection of 0.1 or 0.3. The r19F or r19F_CX4C_ virus in PBS or PBS alone (mock), was added to the apical surface of the cells and incubated for 2 h at 37 °C. The cells were incubated for 2 or 4 days at 37 °C and 5% CO_2_ and then collected with trypsin-0.05% EDTA. Total RNA was extracted and purified from the cells and RNA was reverse transcribed into cDNA and quantitative PCR was carried out with specific primer pairs for the indicated cytokine/chemokine. The C_T_ values were normalized using control β-actin C_T_ values from the same sample. Statistical significance is indicated: *, *p* < 0.05 by Wilcoxon matched pairs test.

**Table 1 vaccines-07-00069-t001:** Vaccination, challenge, and specimen collection schedule.

Group	N	Treatment	Dose (per rat)	Immunizationdays	Route	Primary Infection-Harvest Days	Challenge (PFU/rat) Day 49	Challenge-Harvest Day
Mock	5	PBS	n/a	0, 28	IM	-	PBS	54
A2	5	PBS	n/a	0, 28	IM	-	5 Log10	54
FI-RSV + A2	5	FI-RSV Lot#100	1:100	0, 28	IM	-	5 Log10	54
A2 + A2	5	A2/A2	10^5^ PFU	0	IN	-	5 Log10	54
r19F or r19F + A2	20	r19F or r19F/A2	2 × 10^5^ TCID_50_	0	IN	1,5,49	5 Log10	54
R19F_CX4C_ or R19F_CX4C_ + A2	20	r19F_CX4C_ or r19F_CX4C_/A2	2 × 10^5^ TCID_50_	0	IN	1,5,49	5 Log10	54

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
