# Peer review of "Mutation of Respiratory Syncytial Virus G Protein’s CX3C Motif Attenuates Infection in Cotton Rats and Primary Human Airway Epithelial Cells"

_vaccines, 2019, doi:10.3390/vaccines7030069_

Round 1

Reviewer 1 Report

In this study, the authors describe the infection of cotton rats and primary human airway epithelial cells (pHAECs) with RSV r19F and a mutant virus in which the CX3C motif of the G glycoprotein has been changed to CX4C (r19F/CX4C). The authors showed that the CX4C mutation decreased virus replication and host responses as compared to the r19F (CX3C) virus. Results are also compared to infections with the prototype strain RSV A2.

The study is reasonably well built, and the results are generally supported by the experimental data. However, there are some points for consideration:

1.- While, for the experiments, RSV r19F and  r19F/CX4C were purified, RSV A2 was not. This is an important drawback since it has been shown that non-viral antigens present in unpurified RSV stocks affect the immune response and pathology in the cotton rat model (Shaw et al., 2013). This makes it difficult to compare the results from r19F viruses and A2 infections. All experiments should have been done with purified viruses.

An example, r19F strain has mutations to stabilize F in its pre-fusion conformation, which would induce higher titers of neutralizing antibodies. However, it induced lower levels of neutralizing antibodies than the parental A2. The authors themselves argue that this may be due to differences in virus preparation and inoculum since r19F and r19F/CX4C were purified through a sucrose cushion while the A2 virus was not.

2.- The authors claim that the r19F/CX4C virus induces less pulmonary inflammation in cotton rats. However, only alveolitis was significantly decreased in r19F/CX4C infections when compared to r19F, while peribronchiolitis, perivasculitis, and interstitial pneumonia were not. This result weekly support the authors’ conclusions.

3.- Similarly, only the expression of Muc5ac (but not that of CX3CL1, IL-6, Muc5b, and CCL7)  was decreased in r19F/CX4C infections of pHAEC. In contrast, Muc5b and CCL7 were upregulated. This result should be explained and discussed more extensively. In general, infections of pHAEC do not add new relevant information to that previously reported by the authors and other groups (Chirkova et al., 2013 and 2015; Johnson et al., 2015).

Minor points:

1.- In the introduction, it would be helpful to briefly describe the CX3C (182CWAIC186) and CX4C (182CWAIAC187) motifs.       

 2.- Figure legends should give more information about the experiments. For example, which dilution of sera is represented in Fig. 3A? What MOI was used in Fig. 5 and 6? How many replicates were done to calculate the means and SEMs of Fig. 5 and 6? Cells from all three patients were infected in Fig. 6?

3.- Why are not shown the results of the challenge with RSV A2 (lines 214-218)?.

4.- Fig. 1. Why is A2 not detected in the lung and nose at day 1?

5.- Fig. 3. How do the authors explain that there is no decrease in neutralizing antibody titers at day 49 while anti-F antibodies do decrease?

6.- Lines 290-291: “PCR for RSV RNA showed an average of…” Is this referred to Fig. 6 RSV M?

Author Response

Reviewer 1

1.       While, for the experiments, RSV r19F and  r19F/CX4C were purified, RSV A2 was not. This is an important drawback since it has been shown that nonviral antigens present in unpurified RSV stocks affect the immune response and pathology in the cotton rat model (Shaw et al., 2013). This makes it difficult to compare the results from r19F viruses and A2 infections. All experiments should have been done with purified viruses.

An example, r19F strain has mutations to stabilize F in its pre-fusion conformation, which would induce higher titers of neutralizing antibodies. However, it induced lower levels of neutralizing antibodies than the parental A2. The authors themselves argue that this may be due to differences in virus preparation and inoculum since r19F and r19FCX4C were purified through a sucrose cushion while the A2 virus was not.

Response: Note that the primary comparison is between r19F and r19FCX4C viruses. We feel this is the best comparison for the goal of this study, i.e. to look at the effect of the CX4C mutation in the cotton rat model. In the mouse model, there are significant differences in disease phenotype between the r19F virus (induces IL13, lung mucus, and obstructive airway disease) and the A2 virus (does not induce IL13, lung mucus, and obstructive airway disease).  These differences mean the A2 virus is not the correct choice to assess the effect of the CX4C mutation in r19F.  We included the A2 virus because is the virus that Sigmovir uses in other vaccine studies and it provides a point of comparison to these other studies.  For this purpose, we felt it was important to use the virus preparation they use rather than purified A2.  The available resources did not allow us to include additional groups of cotton rats.

We have revised results on p. 6, lines 219-224 (version with track changes), to better highlight the focus on the comparison between r19F and r19FCX4C.

2.       The authors claim that the r19F/CX4C virus induces less pulmonary inflammation in cotton rats. However, only alveolitis was significantly decreased in r19F/CX4C infections when compared to r19F, while peribronchiolitis, perivasculitis, and interstitial pneumonia were not. This result weekly support the authors’ conclusions.

Response: We have done additional analyses that show the lower inflammation for all parameters for CX4C compared to CX3C infection is significantly different for all parameters combined by ANOVA (Kruskal-Wallis) and two individual parameters are significantly different by Mann-Whitney. These differences are in the results, lines 248-253, and in Figure 2. The new analysis is noted in Figure 2B.  We added pictures of lung histopathology (Figure 2A).

3.       Similarly, only the expression of Muc5ac (but not that of CX3CL1, IL-6, Muc5b, and CCL7)  was decreased in r19F/CX4C infections of pHAEC. In contrast, Muc5b and CCL7 were upregulated. This result should be explained and discussed more extensively. In general, infections of pHAEC do not add new relevant information to that previously reported by the authors and other groups (Chirkova et al., 2013 and 2015; Johnson et al., 2015).

Response. We have expanded our discussion of these measures on lines 344-346 in results and lines 388-394 in the discussion.  We included references with these discussions.

Minor points.

1.       In the introduction, it would be helpful to briefly describe the CX3C (182CWAIC186) and CX4C (182CWAIAC187) motifs.      

Response. As suggested by the reviewer we include the sequences of the CX3C and CX4C viruses, lines 66-67 in the introduction.

2.        Figure legends should give more information about the experiments. For example, which dilution of sera is represented in Fig. 3A? What MOI was used in Fig. 5 and 6? How many replicates were done to calculate the means and SEMs of Fig. 5 and 6? Cells from all three patients were infected in Fig. 6?

3.       Why are not shown the results of the challenge with RSV A2 (lines 214-218)?.

Response.  The data is included in the text, i.e. no virus replication detected in the nose of lung for RSV/A2, r19F or r19FCX4C and titers for FI-RSV and mock given. 

4.       4.- Fig. 1. Why is A2 not detected in the lung and nose at day 1?

Response. We did not test day 1, only day 5, after primary RSV/A2 infection. The figure 1 legend notes this.  We now include NT in the figure 1 to highlight this fact.

5.       Fig. 3. How do the authors explain that there is no decrease in neutralizing antibody titers at day 49 while anti-F antibodies do decrease?

Response. We do not have an explanation for this finding. The response kinetics of different antibodies can vary.  Data from later time points, which we do not have, might help clarify this finding.   

6.       Lines 290-291: “PCR for RSV RNA showed an average of…” Is this referred to Fig. 6 RSV M?

Response.  Yes, this refers to Figure 6.  We now refer to Figure 6 in the text, line 324.

Reviewer 2 Report

Ha et al. studied the impact of a mutation in RSV G protein CX3C-motif (CX4C) in cotton rats and primary human airway epithelial cells using a potential chimeric-RSV-attenuated (r19F-CX4C) vaccine virus. This is a well-written manuscript. However, I have a few major concerns:

Major comments: 

1.  The authors chose to use a chimeric r19F strain of RSV for determining the RSV G protein CX3C-motif mutational-effect on virus attenuation in vitro and in vivo, as the chimeric strain is known to induce higher titer of neutralizing antibodies compared to wild-type A2 strain. In this manuscript, the chimeric strain did not induce higher titer of neutralizing antibodies and not even F antibody titer (Fig 3). The authors even used 2x more inoculum compared to wildtype A2 strain. Additionally, no virus was detected in the lung or nose tissue of previously infected with r19F, r19F-CX3C, or A2 in the challenge study (line 215, data not shown). It suggests sterile immunity regardless of CX3C mutation or strain. The whole manuscript focused CX3C-motif mutational-effect on r19F strain instead of A2 strain. Therefore, the significance and novelty of this study are not identifiable. CX3C-motif mutational-effect in A2 strain and stability of this mutation need to be determined. 

2. Figure 1: RSV A2 titer was not provided or no virus was detected at day 1 post-infection in both lung and nasal tissue. It is not clear whether RSV A2 titer was not detectable at day1 post-infection. It is also not clear whether the r19F and r19F-CX4C virus titer at day 1 post-infection was from the inoculum (used 2x inoculum). Back titration of the virus inoculum would explain.  

3.The authors mentioned in two separate locations about preliminary transcriptome studies (lines 261 and 300) for the identification of cytokines, chemokines, and anti-viral protein (e.g. MX1). Please provide the detail of the study. Relevancy of MX1 expression profile in RSV infection needs to be described.

Minor comments:

1. Please provide representative images for lung histopathology for figure 2. 

2. Figure 6: It is not clear whether RSV A2 mRNA was not detectable or data was not provided, therefore attenuation observed r19F-CX4C compared to r19F not necessarily to A2.  

3. Please discuss the potential of RSV attenuated strain by deleting G for vaccine potential. 

Author Response

Reviewer 2

Ha et al. studied the impact of a mutation in RSV G protein CX3C-motif (CX4C) in cotton rats and primary human airway epithelial cells using a potential chimeric-RSV-attenuated (r19F-CX4C) vaccine virus. This is a well-written manuscript. However, I have a few major concerns:

Major comments: 

1.     The authors chose to use a chimeric r19F strain of RSV for determining the RSV G protein CX3C-motif mutational-effect on virus attenuation in vitro and in vivo, as the chimeric strain is known to induce higher titer of neutralizing antibodies compared to wild-type A2 strain. In this manuscript, the chimeric strain did not induce higher titer of neutralizing antibodies and not even F antibody titer (Fig 3). The authors even used 2x more inoculum compared to wildtype A2 strain. Additionally, no virus was detected in the lung or nose tissue of previously infected with r19F, r19F-CX3C, or A2 in the challenge study (line 215, data not shown). It suggests sterile immunity regardless of CX3C mutation or strain. The whole manuscript focused CX3C-motif mutational-effect on r19F strain instead of A2 strain. Therefore, the significance and novelty of this study are not identifiable. CX3C-motif mutational-effect in A2 strain and stability of this mutation need to be determined. 

Response. As noted in the manuscript, we chose to use the r19F virus to assess the effect of the CX4C mutation in the cotton rat.  The F protein in this virus is partially pre-fusion stabilized, is more stable for storage purposes, and in the mouse model induces higher titer of neutralizing antibodies compared the A2.  These features were attractive for a candidate vaccine.  We do not feel the A2 backbone is the only appropriate one for vaccine development.  We expected that the r19F virus would, in this study in cotton rats, induce lower titers of neutralizing antibodies.  We feel this is in addition to the data on the effect of the CX4C mutation is helps to guide next steps in determining the role of the mutations that modify binding to CX3CR1 in developing an RSV vaccine.  This study, for the first time, assesses the effect of the CX4C mutation in the cotton rat.  Given the challenges in identifying a safe and effective live attenuated RSV vaccine, the potential benefits of this mutation, as show in earlier mouse studies and in this cotton rat study, are substantial.  Note that we have studied the effect of CX4C mutation in the A2 backbone in the mouse model (Boyoglu-Barnum et al. J. Virol. 2017.91: e02059-16) and found markers of lung disease in mice after primary challenge with the A2CX4C and r19fCX4C viruses to be very similar.

Since we had not previously used the r19F virus in the cotton rat, we used a 2X inoculum to add assurance that it would replicate sufficiently well for the study purposes. Note that the focus of the paper is the comparison between the wildtype and CX4C r19F virus and the inoculum titers are the same for these two viruses.  

No virus in lung tissue is expected in re-challenge studies with live RSV. This confirms that the CX4C virus was as effective as the two wild type strains in inducing an immune response that stopped virus replication in this model.  We feel this is expected and a good result for the CX4C virus. 

2.     Figure 1: RSV A2 titer was not provided or no virus was detected at day 1 post-infection in both lung and nasal tissue. It is not clear whether RSV A2 titer was not detectable at day1 post-infection. It is also not clear whether the r19F and r19F-CX4C virus titer at day 1 post-infection was from the inoculum (used 2x inoculum). Back titration of the virus inoculum would explain.  

Response. We did not test day 1, only day 5, after primary RSV/A2 infection. The figure 1 legend notes this.  We now include NT in the figure 1 to highlight this fact.  We did not do a back titration of the virus inoculum. We suspect that non-replicating RSV loses titer in the lung over 24 hours and residual inoculum, therefore, not likely to affect the lung titer on day 1. It is clear, as indicated by the increase in lung titer on day 5, that RSV replicates in the lung of the cotton rat. 

3.     The authors mentioned in two separate locations about preliminary transcriptome studies (lines 261 and 300) for the identification of cytokines, chemokines, and anti-viral protein (e.g. MX1). Please provide the detail of the study. Relevancy of MX1 expression profile in RSV infection needs to be described.

Response. We have added additional information and references related to the relevancy of MX1 expression in RSV infection lines 298-305 in results.

Minor comments:

1.     Please provide representative images for lung histopathology for figure 2.

Response. We now include lung representative histopathology figures, Figure 2A

2.     Figure 6: It is not clear whether RSV A2 mRNA was not detectable or data was not provided, therefore attenuation observed r19F-CX4C compared to r19F not necessarily to A2.  

Response.  We did not include the A2 virus in these pHAEC experiments.  Since the focus of the paper is the effect of mutating the CX3C motif to CX4C and since A2 and r19F virus have different disease manifestations, we feel the r19FCX4C virus should be compared to r19F. We have published data on A2 compared to A2CX4C in pHAECs, Chirkova et al. 2015

3.     Please discuss the potential of RSV attenuated strain by deleting G for vaccine potential. 

Response.  Deletion of the G protein does attenuate RSV. However, data in mice suggest that G- induced immune responses add to protection, both through an antiviral and anti-inflammatory response.  We now note the advantages of maintaining a G antibody response in the last sentence in the discussion. 

Round 2

Reviewer 2 Report

Major comments:

The authors used r19F virus to assess the effect of the CX4C mutation in the cotton rat and they included infection with the A2 virus to provide a comparison to other cotton rat studies. It is not clear how A2 strain data in this manuscript compare results to other cotton rat studies. The authors’ response to this reviewer’s minor comments 2 also suggests A2 data is irrelevant. The A2 strain data need to be removed from this manuscript. When this reviewer agrees that A2 strain is not the only backbone for RSV vaccine development, r19F virus infection in the cotton rat has not been explored enough before using for characterization of CX4C mutational effect. 

The CX4C mutational effect on r19F virus was not substantial e.g. Fig 3, 5, and 6.  

Minor comments:

Please provide a note or reference in the manuscript for reader’s convenience about the authors' response “We suspect that non-replicating RSV loses titer in the lung over 24 hours and residual inoculum, therefore, not likely to affect the lung titer on day 1”. 

Line 332 please provide the reference. 

Author Response

Reviewer 2

Major comments:

1.     The authors used r19F virus to assess the effect of the CX4C mutation in the cotton rat and they included infection with the A2 virus to provide a comparison to other cotton rat studies. It is not clear how A2 strain data in this manuscript compare results to other cotton rat studies. The authors’ response to this reviewer’s minor comments 2 also suggests A2 data is irrelevant. The A2 strain data need to be removed from this manuscript.

Response. We agree with the reviewer that the data with the A2 strain is not helpful in the comparison of the r19F and r19FCX4C viruses and this raise the possibility of excluding the A2 data.  However, it does provides a reference point for comparing results of this study with other studies in cotton rats (some are now referenced in the article in lines 216-217) and another comparison for the FI-RSV data. A more important rationale, we feel, is the comparison between the lung inflammation with r19FCX4C primary infection and A2 primary infection. This comparison suggests that a candidate, live attenuated vaccine strain with a CX3C-related mutation will need additional attenuating mutations. Though, we agree with the reviewer that the A2 data does not inform the comparison between the r19F and r19FCX4C viruses, we feel it does add sufficient value to merit inclusion in the paper.

2.     When this reviewer agrees that A2 strain is not the only backbone for RSV vaccine development, r19F virus infection in the cotton rat has not been explored enough before using for characterization of CX4C mutational effect. 

Response. Note that the r19F virus is on the A2 backbone just with a different F protein. There are several publications that include r19F virus infection in the cotton rat (reference in lines 216-217) though there are none with r19FCX4C virus. Cotton rat studies of candidate live attenuated RSV strains such as the r19FCX4C virus strains are done without extensive testing in the cotton rat. Though additional studies might be of interest, we feel this study is consistent with other cotton rat studies of candidate vaccine viruses and provides useful information on the r19FCX4C virus as a candidate vaccine.

3.     The CX4C mutational effect on r19F virus was not substantial e.g. Fig 3, 5, and 6.

Response. The purpose of the study is to address the question, “What is the effect of the CX4C mutation on virus replication and disease/host response in the cotton rat and pHAECs?” We think the studies presented inform this question independent of the degree of differences seen. We do, however, think many of the significant differences between the r19F virus and r19FCX4C virus are noteworthy including virus replication on day 5 after primary infection in the cotton rat, Fig 1; the differences in interstitial pneumonia and alveolitis after primary infection in the lung, Fig 2; the differences in MX1, TNF-α, INF-γ, and MCP-1 on day 5 after primary infection, Fig 4; the differences in RSV+ cells for all days in pHAECs, Fig 5; and the differences in MUC5AC and CCL7 on day 4 post infection of pHAECs. We would not expect much difference in these measures in the re-challenge studies in the cotton rat since there is little or no virus replication in challenged animals previously infected with RSV. We have revised lines 357-362 to highlight that we are focusing on differences with primary infection in the cotton rat, identify which differences we feel are noteworthy, and note why we would not expect differences in re-challenge studies. How substantive these differences are to infection in humans, of course, will have to be determined in clinical trials. In other parts of the discussion, we note differences we feel are most important in pHAECs.

Minor comments:

1.     Please provide a note or reference in the manuscript for reader’s convenience about the authors' response “We suspect that non-replicating RSV loses titer in the lung over 24 hours and residual inoculum, therefore, not likely to affect the lung titer on day 1”. 

Response.  We have added a comment and a reference as requested in lines 222-223 in the revised manuscript.

2.     Line 332 please provide the reference.

Response.  Reference added, line 336 in the revised manuscript.

Round 3

Reviewer 2 Report

The authors have provided a justification for using A2 strain data in the manuscript. However, it needs to be clearly mentioned in the manuscript. Additionally, please elaborate or discuss any limitations in the manuscript e.g. why r19F virus did not produce higher neutralizing antibody titer. 

Author Response

Reviewer 2

1.     The authors have provided a justification for using A2 strain data in the manuscript. However, it needs to be clearly mentioned in the manuscript.

Response.  We have expanded the rationale for including the A2 virus in this study in lines 216-219

2.     Additionally, please elaborate or discuss any limitations in the manuscript e.g. why r19F virus did not produce higher neutralizing antibody titer. 

Response.  We note this limitation more clearly in lines 387-403 and note other limitations in the discussion including the possibility that the 19FCX4C virus is not sufficiently attenuated in lines 358-371.
